# Longitudinal relations among inattention, working memory, and academic achievement: testing mediation and the moderating role of gender

Sarah A. Gray[1], Maria Rogers[2], Rhonda Martinussen[1] and Rosemary Tannock[1,3]

[1] Ontario Institute for Studies in Education, University of Toronto, Canada
[2] University of Ottawa, Canada
[3] Research Institute of the Hospital for Sick Children, Canada

## ABSTRACT

**Introduction.** Behavioral inattention, working memory (WM), and academic achievement share significant variance, but the direction of relationships across development is unknown. The aim of the present study was to determine whether WM mediates the pathway between inattentive behaviour and subsequent academic outcomes.

**Methods.** 204 students from grades 1–4 (49.5% female) were recruited from elementary schools. Participants received assessments of WM and achievement at baseline and one year later. WM measures included a visual-spatial storage task and auditory-verbal storage and manipulation tasks. Teachers completed the SWAN behaviour rating scale both years. Mediation analysis with PROCESS (*Hayes, 2013*) was used to determine mediation pathways.

**Results.** Teacher-rated inattention indirectly influenced math addition fluency, subtraction fluency and calculation scores through its effect on visual-spatial WM, only for boys. There was a direct relationship between inattention and math outcomes one year later for girls and boys. Children who displayed better attention had higher WM scores, and children with higher WM scores had stronger scores on math outcomes. Bias-corrected bootstrap confidence intervals for the indirect effects were entirely below zero for boys, for the three math outcomes. WM did not mediate the direct relationship between inattention and reading scores.

**Discussion.** Findings identify inattention and WM as longitudinal predictors for math addition and subtraction fluency and math calculation outcomes one year later, with visual-spatial WM as a significant mediator for boys. Results highlight the close relationship between inattention and WM and their importance in the development of math skills.

Corresponding author
Sarah A. Gray,
Sa.gray@mail.utoronto.ca

A strong body of literature has provided evidence of a link between inattentive behavior in the classroom and academic underachievement (for a review, see *Polderman et al.,*

*2010*). As a behavioural descriptor, attention refers to overt on-task behaviours (for example, visual fixation on a relevant stimulus such as a teacher) as well as organization (for example, keeping track of materials) and is measured using behavioural rating scales (for example, the SWAN rating scale, Conners-3) filled out by a parent or teacher who regularly observes a child's behaviour. Inattention refers to off-task behavior, and includes the concept of disorganization. Although Polderman and colleagues included the behavioural dimension of hyperactivity in their review, it is the dimension of inattention that has consistently been found to be a risk factor for poor academic achievement across development (*Garner et al., 2014*; *Pingault et al., 2011*). Teacher-ratings of behavioural inattention are more strongly linked to academic outcomes than are parent-ratings, and are more sensitive to the demands of the classroom environment (*Garner et al., 2014*).

**Inattention and math achievement.** Teacher-rated inattention is an independent predictor of performance in multiple achievement domains that are important throughout the elementary school years, including arithmetic fluency (*Fuchs et al., 2006*; *Lewandowski et al., 2007*), arithmetic word problems (*Fuchs et al., 2006*; *Swanson, 2011*) and algorithmic computation (*Fuchs et al., 2006*; *Li & Geary, 2013*; *Raghubar et al., 2009*), as well as for composites of arithmetic fluency and algorithmic computation (*Fitzpatrick & Pagani, 2013*; *Gold et al., 2013*). These three domains of math are distinguishable (*Fuchs et al., 2006*). Arithmetic fluency is defined as solving simple math facts, with a timing component, where students are expected to quickly and accurately solve math fact problems. As children become efficient counters, associations between pairs of numbers become consolidated in long-term memory, therefore relying more on retrieval memory and putting less burden on working memory (WM) for answering math fact questions fluently (*Geary, Brown & Samaranayake, 1991*). Petrill and colleagues (*2012*) found that arithmetic fluency is genetically distinct from other non-timed measures of math calculation, problem solving and number concepts. Arithmetic fluency plays a role in the development of algorithmic computation, which is defined by *Fuchs et al.* (*2006*, p.30) as "adding, subtracting, multiplying or dividing whole numbers, decimals or factions using algorithms and arithmetic." This type of math is differentiated from arithmetic fluency as it includes carrying and borrowing; moving outside of simple arithmetic to include the use of algorithm, and necessitating the ability to follow procedural steps as well as reliance on math fact retrieval.

**Inattention and reading achievement.** Reading fluency is another important domain of achievement during the elementary school years, as it is a consistent predictor of later reading comprehension skills (*Pearce & Gayle, 2009*; *Roehrig et al., 2008*). The ability to read fluently in the early grades is also predictive of high-stakes achievement test scores in elementary and middle school, and continues to predict reading comprehension scores into adulthood (*Baker et al., 2014*; *Tighe & Schatschneider, 2014*). There is some evidence that reading fluency is linked to attention, in that inattentive behavior is a predictor of poor reading fluency outcomes in typical developing school children (*Pham, 2013*). A study using a community sample of elementary school children found that mid-term teacher-rated inattention predicted word reading fluency at the end of the same year,

although it did not predict basic reading (word reading without timed component and decoding ability) (*Grills-Taquechel et al., 2013*). Studies with clinical groups have also found that children with Attention-Deficit/Hyperactivity Disorder (ADHD) have lower reading fluency outcomes than their peers (*Jacobson et al., 2011*; *Willcutt et al., 2007*).

The mechanisms of association between inattention and math and reading outcomes are not yet delineated. Although it is teacher-rated inattention that is strongly linked with poor academic achievement, attention encompasses both behavioural and cognitive components, and these two aspects of attention do not readily map onto each other (for a review, see *Tannock, 2003*). As a cognitive descriptor, attention refers to a complex set of processes that operate through a series of neural networks. Three specific networks have been delineated: alerting, orienting and executive control (*Posner & Rothbart, 2007*). It is not known whether cognitive aspects of attention mediate the relationship between behavioural inattention and poor academic achievement. Within the cognitive network of executive control, the functional domain of working memory (WM) has been implicated in math and reading achievement and is strongly related to inattention, and thus presents as a possible mediating variable within this relationship (*Fuchs et al., 2005*; *Martinussen & Tannock, 2006*; *Swanson & Beebe-Frankenberger, 2004*).

A number of educational studies draw upon Baddeley's multicomponent model of WM in which WM is viewed as a limited-capacity system that temporarily holds and manipulates information. This model includes separate storage modules for auditory-verbal (phonological loop) and visual-spatial information (visual-spatial sketchpad), and a central executive component that interfaces with other systems such as long term memory and perceptual systems (*Baddeley, 2003*; *Baddeley, 2012*). Within this model, both short-term storage modules and modules that process or manipulate information are considered to be part of WM. Following from this model, as well as research that provides evidence that these domains overlap significantly and may not tap into separate constructs, both storage and manipulation components will be conceptualized under the domain of WM within the current study (*Colom et al., 2006*; *Engle et al., 1999*; *Miyake et al., 2001*). Differences have been found, however, between tasks that require short-term storage or manipulation, with the latter showing relationships to fluid intelligence and cognitive aptitudes (*Cowan, 2008*; *Engle et al., 1999*).

Children with poor WM ability demonstrate impaired academic performance, including impaired performance on tests of overall reading and math, and reading fluency (*Alloway et al., 2009*; *Bental & Tirosh, 2007*; *Gathercole & Pickering, 2000*; *Jacobson et al., 2011*). These same children are rated by teachers as having more problems with inattention and distractibility (*Alloway et al., 2009*). Similarly, a study with a sample of children representing the normal range of WM ability found that WM is long-term predictor of literacy and numeracy outcomes (*Alloway, Elliott & Place, 2010*). A recent study found that WM (a composite of both auditory-verbal and visual-spatial WM), was an important predictor of math achievement for students with high levels of ADHD symptoms (*Rennie, Beebe-Frankenberger & Swanson, 2014*). Visual-spatial storage, when measured in pre-school children, was also found to predict first grade math outcomes (*Bull, Espy*

 

*& Wiebe, 2008*). Working memory deficits often co-occur with attention difficulties, both in those individuals with disorders of attention and across the spectrum of typical behaviour (*Gathercole & Pickering, 2000*; *Martinussen & Tannock, 2006*; *Willcutt et al., 2005*). Moreover, when examined across one school year, inattention, WM and academic fluency were found to share a significant amount of variance in a community sample of elementary school children (S Gray et al., 2014, unpublished data), supporting the hypothesis that these three factors comprise a triad of impairment during the elementary years and into high-school (*Martinussen et al., 2005*; *Rogers et al., 2011*).

Currently there is no robust evidence regarding the direction of the relationships within this triad of impairment, and causal pathways are unknown. One study found that trajectories of ADHD behaviour could be established based on cognitive features at 15 and 24 months, and that those with more severe ADHD symptoms in grade 3 did show some behavioural differences prior to starting school (*Arnett, Macdonald & Pennington, 2013*). The researchers found that early signs of both behavioural and cognitive difficulties were associated with a stable trajectory of poor academic achievement into grade 3 (*Arnett, Macdonald & Pennington, 2013*). Although this study provides evidence as to the early emergence of both behavioural and cognitive difficulties, and their association with low academic achievement, a grouping of cognitive features based on general intelligence and grouping inattention with an 'externalizing behaviour' composite does not allow for looking at domains of specific relevance to academic achievement, such as WM and the spectrum of inattention. Another study, examining a sample of term and pre-term children, found that a measure of executive function (EF) (including visual-spatial WM) did not contribute unique variance to teacher-rated inattention scores in preschool, but visual-spatial span did contribute unique variance to these scores in primary school (*Aarnoudse-Moens et al., 2013*). These studies indicate changes in the relationship between teacher-rated inattention and WM throughout early development.

Other studies have investigated possible mediators that provide some account of the consistent relationship between inattention and academic achievement. In a sample of high school students presenting with clinical and sub-clinical levels of ADHD symptoms, WM was found to be a mediator of the relationship between inattention and reading and math composite scores (*Rogers et al., 2011*). *Thorell (2007)* examined WM in a mediating role within an EF composite score. They found that this EF score mediated the relationship between inattention and pre-academic skills in kindergarten-aged children (*Thorell, 2007*).

The current study sought to extend these studies to a community sample of elementary school children and to further delineate the nature of the relationship between classroom inattention, WM domains and academic achievement through using a longitudinal mediation design. Differential influences of visual-spatial and auditory-verbal WM are of interest, given previous research that implicates visual-spatial WM as an important factor in math achievement in elementary and high school, and previous findings of differential relationships between WM domain and achievement domain (*Li & Geary, 2013*; *Rogers et al., 2011*). As sex differences are often evident in overall levels of inattentive behavior (*Gershon, 2002*), sex was investigated in the current study. Moreover, given recent, although

limited evidence that sex differences in attention disorders may be due to underlying genetic and cognitive differences between the sexes (*Arnett et al., 2014*), we included sex as a moderator of the direct and indirect effects. Based on the previous studies described, as well as on examination of this sample within a one year time frame (S Gray et al., 2014, unpublished data), it is hypothesized that there will be a direct relationship between classroom inattention at one point in time and both math and reading outcomes one year later. Additional hypotheses posit that inattention will indirectly influence math outcomes through visual-spatial and auditory-verbal WM, and indirectly influence reading outcomes through auditory-verbal WM.

## MATERIALS AND METHODS

### Participants

Participants were 204 elementary school-aged children (49.5% female) in grades 1–4 (ages 5–9, $M = 7.67$, $SD = 0.91$), who were drawn from a larger sample of 524 students, as described below. Students and their teachers and parents were recruited from a large suburban and rural school district in Southern Ontario, Canada. The 7 participating schools (20% of the 33 schools in the district) were stratified across socio-economic groups. Stratifying for sex, this subsample of 204 was created by taking 2–3 students in each class from the highest, middle and lowest ranking levels of attention, based on teacher ratings of inattentive behaviour in the classroom, which were rank ordered. This smaller sample, representative of the continuum of attention across students, was then given more in-depth academic and cognitive assessments in the second half of each study year.

The majority of participants were Caucasian (80.6%) with English as their primary language (83.3%). All students that were in mainstream English or French classrooms (29% in French Immersion) were eligible for the study, providing that they did not have major sensory or motor impairment that would preclude the ability to complete the tasks or hear instructions. Teacher reports indicated that 11.8% of sample had an Individual Education Plan (IEP) with 5.5% identified with ADHD, 3.8% a learning disability, 4.9% a language impairment, 1.6% a behaviour difficulty, 0.5% a developmental disability. In terms of parent education level, 2.7% of participating parents had less than high school education, 5.5% had graduated from high school or an equivalent, 57.7% had graduated from college or university, and 11% had a post graduate degree. No sex differences were found on any demographic variables, with one exception: females were more likely to have a parent (92.3% of parents who filled out questionnaires were mothers, and 7.7% were fathers) with less than high school education.

### Procedures

In accordance with procedures approved by the hospital and school board Institutional Review Boards (REB approval number 1000013136), study information was presented in an initial meeting with principals of potential participant schools. Interested principals then contacted the research team, after which an information session for teachers was held at each participating school.
In the Canadian school system, the school year starts in September and ends in June. Thus, November is in term one (term A) and April is in term two (term B) of the school year. In the current study, there were four waves of data collection across two years. The first wave took place in November of study Year 1, and will be referred to as Year 1 term A (Year 1A). The second wave took place in April of Year 1 and will be referred to as Year 1 term B (Year 1B). Similarly, in the second year of the study, data was collected in November (Year 2A) and in April (Year 2B).

Teachers and parents who gave written informed consent to participate in the current study completed questionnaire packages in November of Years 1 and 2 of the study (Year 1A, Year 2A). This gave the teachers two months to get to know their students (September and October) before completing the questionnaires. At the time of consent, parents were aware that their children might participate in either two or four testing sessions across the two years. Children who had written informed consent from parents and gave verbal assent, participated in academic testing sessions in November of Years 1 and 2 of the study (Year 1A, Year 2A). All assessments were conducted in English, and all materials were English. As described above, after the teacher-rated inattentive behaviour questionnaires were completed, a subset of students from each class, from the lowest, middle and highest bracket of the continuum of attention were selected to participate in further tests of cognitive (including working memory) and academic functioning. These further tests were administered to the same subset of 204 students in April of study Years 1 and 2 (Year 1B, Year 2B).

## Measures

The following measures, including a behaviour questionnaire, and standardized tests of academic achievement and WM were selected from a larger study that included a range of behavioural, cognitive and academic measures.

### *Assessment of classroom attention*

Classroom attention was measured using the *Strengths and Weaknesses of Attention-Deficit/Hyperactivity Disorder Symptoms and Normal Behaviour Scale (SWAN; http://www. adhd.net/SWAN_SCALE.pdf),* completed by teachers in November of Year 1 and Year 2 (Year 1A, Year 2A) of the study. This scale assesses behaviour using a sensitive 7-point scale (3 = Far below average, 2 = Below average, 1 = Slightly below average, 0 = Average, −1 = Slightly above average, −2 = Above average, −3 = Far above average). This allows for measuring the full range of behavioural attention in a population-based sample, using positively worded probes. This design avoids psychometric flaws such as negative or positive skewedness that can arise with 4 point-scales, in which scores for the majority of children cluster around zero. The SWAN scale provides a range of scores for children who have average attention as well as good or poor attention at either end of the spectrum (*Arnett et al., 2011*). The scale is divided into 'inattention' and 'hyperactivity' subscales. The inattention subscale only was employed in this study, considering the large body of evidence that links inattention with academic achievement, and does not provide evidence of such a link between hyperactivity and academic outcomes (*Garner et al., 2014*; *Rabiner*

*& Coie, 2000*). This inattention subscale consists of 9 items. Internal consistency of this scale is acceptable and consistent with other often used behaviour rating scales (full scale, $\alpha = .88$; inattention subscale, $\alpha = .94$. Test-retest reliability estimates for the full scale range from .72–.90 *Arnett et al., 2011*; *Swanson et al., 2001*; *Young et al., 2009*). In the current sample, a correlation of .74 was found for the inattention subscale at Year 1A and Year 2A. Scores are distributed and coded based on a 7-point scale: Negative scores indicate stronger attention; lower levels of inattentive behaviour, while positive scores indicate weak attention; higher levels of inattentive behaviour.

### Measures of math achievement

To assess students' math abilities across the two years, subtests from two commonly used batteries were administered at each wave of data collection. The addition and subtraction probes from *AIMSweb® M-CBM, Mathematics Curriculum-Based Measurement* was used to test grade-level fluency in addition and subtraction, therefore was used as a measure of arithmetic fluency. This reliable and valid curriculum-based measure (CBM) assesses math fluency; probes are taken from the school curriculum and standardized. Test-retest reliability is high (.87), as is inter-rater reliability (.83), and alternate form reliability is moderate (.66) (*Thurber, Shinn & Smolkowski, 2002*). Forms for grades 1–3 included 60 math fact problems (basic subtraction and addition), and forms for grade 4 students included 84 math fact problems. Math problems did not require borrowing or carrying, and contained digits 0–12, thus some computations were either single ($1 + 8$) multi-digit ($11 - 8$). The content was the same for both forms, with the only difference being the number of available questions. The scoring is unique in that credit is given to each individual correct digit that appears in the solution. This allows for a more precise analysis of a child's math skills, as it captures emerging and partial skills as well as fully mastered skills, thus providing a sensitive measure of math fluency. The test is administered in a group format, and students are given 2 min to complete as many problems as they can. This task is sensitive to both short-term and long-term improvement in student achievement, thus is appropriate for a longitudinal study design (*Thurber, Shinn & Smolkowski, 2002*).

The Math Calculation subtest from the *Woodcock-Johnson - III Tests of Achievement (WJ-IIIACH)* was administered to assess a second component of math achievement, within the area of algorithmic computation. The WJ-IIIACH is a highly reliable standardized battery that can be used throughout the academic trajectory. Internal consistency reliability is .86 for Math Calculation (*Woodcock, McGrew & Mather, 2001*). The problems in this task do start out with simple arithmetic (i.e., $2 + 3$), similar to the CBM addition and subtraction fluency tasks, however problems quickly move into borrowing and carrying (i.e., $16 + 6$) and to multiplication and division. The difficulty of the questions increases as the student progresses. The test is timed, however time is not emphasized in the instructions and a full seven minutes is given for the participants to complete as many questions as they can. Partial points are not given as in the CBM tasks, the questions in this task are either given one point or zero points.

Throughout this paper, the WJ-IIIACH Math Calculation task that taps into components of algorithmic computation will be referred to as 'math calculation.' The CBM math

addition and subtraction fluency tasks described above will be referred to as 'addition fluency' and 'subtraction fluency.'

Previous research has found that addition and subtraction fluency (short time limit for simple arithmetic problems) and math calculation (algorithmic computation, longer time limit), cluster together under the narrow math ability factor, and both are related to perceptual speed. However, these domains of math are separable factors, and only math fluency is related to the broad ability of processing speed (*Woodcock, McGrew & Mather, 2001*).

### Measures of reading achievement

Reading fluency was assessed using the *Dynamic Indicators of Basic Early Literacy Skills (DIBELS, 5th ed)*, Oral Reading Fluency Subtest. This test is an individually administered curriculum-based measure (CBM) of oral reading fluency, with good predictive validity, and strong concurrent validity (.91–.96), and alternate-form reliability (.89–.96) (*Good et al., 2001*). Students are given 3 grade level passages to read out loud, and are instructed to read as accurately as possible, and to read as many words as they can within one minute. Points are deducted for omissions, substitutions, inaccurate pronunciation and hesitations over 3 s. The median number of errors across the three passages is scored, as is the median number of correct words; this latter score was used as the oral reading fluency measure in the current study. One subtest from the *Woodcock-Johnson-III Tests of Achievement (WJ-IIIACH)*, Letter-Word Identification, was used from the "Reading ability" cluster of this battery in order to test fluent single word reading ability. This subtest presents single words listed on a page and words increase in difficulty as the student progresses. Credit is given if the word is said out loud smoothly and accurately. Throughout this paper, the CBM Oral Reading Fluency subtest will be referred to as 'Reading Fluency' and the WJ-ACHIII Letter-Word-Identification test will be referred to as 'Word Reading.'

### Assessment of working memory

To assess WM, the following two tests were chosen for their strong psychometric properties and in order to extend previous findings from studies using these measures. The *Wechsler Intelligence Scale for Children (WISC-IV), Digit Span Subtests (DS)* is a widely-used test of auditory-verbal WM, with an internal consistency of .87 and test-retest reliability of .82 (*Wechsler, 2003*). The test requires participants to listen to and recall a series of digits. In the Digit Span Forward task, participants are asked to recall the digits exactly as heard, while in the Digit Span Backward task, participants are asked reproduce the digits heard in backward sequence. The standardized composite score of these two tasks was used in the current study. The *Wide Range Assessment of Memory and Learning (WRAML-2), Finger Windows Forward Subtest (FWF)* was administered in order to assess visual-spatial WM. This test has high internal consistency (.99) and taps into the visual-spatial storage component of WM (*Sheslow & Adams, 2003*). This battery does not contain a visual-spatial WM task that requires manipulation in addition to short-term storage. Participants are presented with an 8 × 11 plastic grid with 'windows' distributed throughout the grid. Participants, who are seated directly across from the examiner, are asked to replicate the

**Peer**J

examiner's visual sequence, created with a pencil tapping different sequences of 'windows.' The sequence becomes longer as participants progress. A standardized score is calculated from the total number of correct sequences that the participant is able to replicate.

## Statistical approach

Missing data was imputed according to the methods suggested by *McKnight et al. (2007)*, when not more than 10–15% of data is missing. It should be noted that for analyses in which parental education is a covariate, only 182 records were complete with this information, thus analyses are based on this reduced sample size. No significant outliers were detected. Assumptions of normality and homoscedasticity were satisfied, with the exception of the Year 2B Math Addition variable, where the Levene's test was significant for the male sample (however, the test was not significant for the full sample). As a precaution, the HC3 test in the SPSS macro PROCESS (*Preacher & Hayes, 2008*) was used to produce heteroscedasticity-consistent standard error estimates for this variable.

### Relationships between study variables

Partial correlations were calculated to examine the relationship between all study variables. Age was placed as a control variable, as it is an important factor in CBMs across grades and initial analysis indicated that age was differentially related to WM variables. A one-way ANOVA was used to examine sex differences between study variables.

### Mediation analyses

All mediation models were designed with visual-spatial WM and auditory-verbal WM at Year 1B as parallel mediators between teacher-rated inattention at Year 1A and academic outcomes at Year 2B, with sex as a moderator. Moderated mediation analyses were carried out using the PROCESS macro for SPSS, developed and described by *Hayes (2013)* and *Preacher & Hayes (2008)*. Their suggested procedures allow for detecting the difference between the direct effect of a predictor on an outcome variable, and the indirect effect after accounting for the mediator. Using this macro also allows for testing the relative strength of auditory-verbal WM and visual-spatial WM as mediators within each analysis (*Hayes, 2013*; *Preacher & Hayes, 2008*). Year 1A academic scores were added in each analysis as a covariate. This model allows for partialling out the influence of baseline academic scores collected at Year 1A and examining influences of each variable across time. Outcome variables were examined in separate models instead of in one simultaneous model in order to elucidate the role of inattention and WM in the development of specific skills within math and reading at the elementary school level. All analyses were carried out with IBM SPSS version 21.

## RESULTS

### Correlations between study variables

Partial correlations between all study variables, controlling for age, are presented in Table 1. Teacher-rated inattention, measured at Year 1A, was significantly correlated in the expected direction with WM measures at Year 1B and all academic outcome variables at Year 2B. All main study variables were significantly correlated in the expected direction at the .01 level,

**Table 1 Correlations table.** Partial correlations, controlling for age, between study variables for full sample ($N = 204$).

| Variables | 1 | 2 | 3 | 4 | 5 | 6 | 7 | 8 | 9 |
|---|---|---|---|---|---|---|---|---|---|
| 1. Teacher-rated inattention | – | | | | | | | | |
| 2. Word reading | −.49[**] | – | | | | | | | |
| 3. Reading fluency | −.54[**] | .81[**] | – | | | | | | |
| 4. Math calculation | −.48[**] | .49[**] | .51[**] | – | | | | | |
| 5. Addition fluency | −.48[**] | .42[**] | .52[**] | .59[**] | – | | | | |
| 6. Subtraction fluency | −.40[**] | .44[**] | .47[**] | .58[**] | .82[**] | – | | | |
| 7. Auditory-verbal WM | −.23[**] | .31[**] | .35[**] | .33[**] | .30[**] | .34[**] | – | | |
| 8. Visual-spatial WM | −.34[**] | .24[**] | .24[**] | .33[**] | .33[**] | .32[**] | .14 | – | |
| 8. Sex | −.37[**] | .08 | .16[*] | .08 | .00 | −.15[**] | −.05 | .16[*] | – |
| 8. Parent education level | −.22[**] | .23[**] | .31[**] | .22[**] | .20[**] | .24[**] | .16[*] | .10 | .03 |

**Notes.**

Significant correlation:

[*] $p < .05$.

[**] $p < .01$.

Measured at Year 1 Time A: Teacher-rated Inattention, Age, Sex, Parent education level.
Measured at Year 1 Time B: All working memory measures.
Measured at Year 2 Time B: All academic measures.
Teacher-rated Inattention, Addition and Subtraction Fluency: total raw scores. Math Calculation, Reading Fluency, Word Reading, working memory variables: Standard Scores.

with the exception of visual-spatial and auditory-verbal WM which were not significantly correlated. This finding is somewhat unexpected as a previous study using an adolescent sample found that both domains of WM were moderately correlated (*Rogers et al., 2011*). This finding is discussed further below.

A very strong positive relationship was found between reading fluency and word reading, and between the math addition and subtraction fluency tests, as would be expected given previous research looking at the overlap and differences within these domains of reading achievement (*Woodcock, McGrew & Mather, 2001*), and because the math CBM subtests are both measures of math fact fluency within the same test format. The correlation between math calculation and fluency of math addition and subtraction was lower by comparison, but still strong. This is also expected, as these domains clearly require a similar skill base, however, math fluency is uniquely related to processing speed, while the calculation subtest is related to perceptual speed, and requires higher level procedural skill (*Woodcock, McGrew & Mather, 2001*). Parent education was weakly to moderately correlated with all variables, with the exception of visual-spatial WM, which appears to be related to sex but not to parental education in this sample. Conversely, auditory-verbal WM was not related to sex, but was weakly correlated with parental education. Sex was significantly correlated with inattention at the .01 level, and with reading fluency, math subtraction and visual-spatial WM at the .05 level. No significant differences on any study variables were found between students in French Immersion or English classrooms ($ps > .05$). For all models, parent education and age were entered into the model as covariates along with Y1A academic scores.

**Table 2   Means and standard deviations for study variables, for girls ($n = 101$) and boys ($n = 103$).**

|  | Total M (SD) | Males M (SD) | Females M (SD) | $F$[a] |
|---|---|---|---|---|
| Teacher-rated inattention | −1.58 (13.92) | 3.43 (13.07) | −6.69 (12.91) | 30.95[**] |
| Word reading | 102.0 (11.79) | 100.53 (12.52) | 103.56 (10.84) | 3.41 |
| Reading fluency | 102.2 (44.26) | 95.95 (44.83) | 108.47 (42.98) | 4.14[*] |
| Math calculation | 95.00 (9.61) | 94.15 (9.35) | 95.87 (9.83) | 1.65 |
| Addition fluency | 41.67 (18.14) | 42.37 (19.51) | 40.96 (16.70) | 0.31 |
| Subtraction fluency | 29.82 (13.90) | 32.23 (14.88) | 27.36 (12.43) | 6.44[*] |
| Auditory-verbal WM | 8.77 (2.45) | 8.78 (2.33) | 8.76 (2.58) | 0.002 |
| Visual-spatial WM | 8.12 (3.25) | 7.73 (3.31) | 8.51 (3.16) | 3.01 |

**Notes.**

[*] $p < .05$

[**] $p < .001$

[a] F statistic for one-way ANOVA.

Measured at Year 1 Time A: Teacher-rated Inattention, Age, Sex, Parent education level.
Measured at Year 1 Time B: All working memory measures.
Measured at Year 2 Time B: All academic measures.
Teacher-rated Inattention, Addition and Subtraction Fluency: total raw scores. Math Calculation, Reading Fluency, Word Reading, working memory variables: Standard Scores.

To further examine sex differences in this sample, means and standard deviations for study variables, for boys and girls are presented in Table 2. Significant sex differences were found for teacher-rated inattention (boys are rated as more inattentive than girls), reading fluency (girls have higher scores than boys), and math subtraction (boys have higher scores than girls). However, when applying a Bonferroni family-wise correction for each of the comparisons, with the threshold for significance at an alpha level of .0063, the only remaining sex difference was between teacher-rated inattention. The eta-squared for this significant difference is 0.13, therefore 13% of the variance in teacher-rated inattention is accounted for by sex. To assess sex differences in the hypothesized model, sex was entered as a moderator of the mediation analyses.

## Mediation analyses

### Math outcomes

Moderated mediation analyses with two parallel mediators, and sex as a moderator, conducted using ordinary least squares (OLS) path analysis, revealed that teacher-rated inattention indirectly influenced math addition and subtraction fluency outcomes through its effect on visual-spatial WM (see Figs. 1 and 2), but for boys only. Unstandardized regression coefficients are reported below and in the figures, in accordance with the recommendation of *Hayes (2013)*. Standardized regression coefficients are not produced by PROCESS, and the absolute size of the direct and indirect effects does not indicate whether effects are small or large, as they are tied to our measures that differ across questionnaire/test. A discussion of effect size follows the presentation of results.

Figure 1 presents results of the analysis with addition fluency as the outcome measure. Children who displayed lower levels of teacher-rated inattention at Year 1A (negative scores correspond to better attention) had higher visual-spatial WM scores ($a = −0.16$,

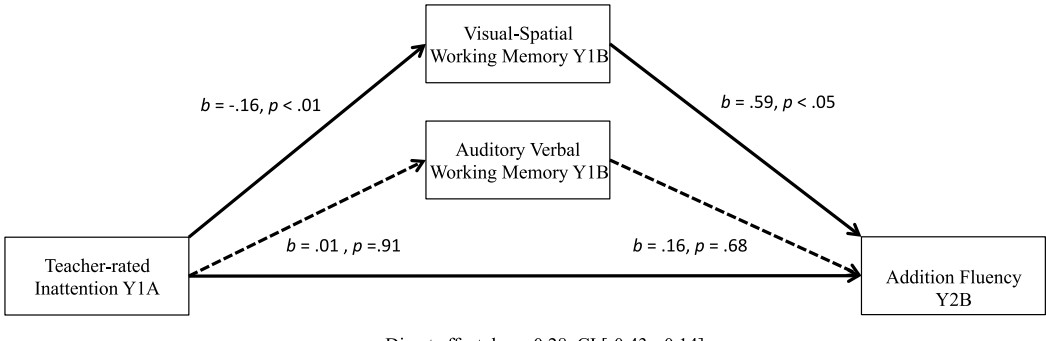

**Figure 1 Significant moderated mediation: visual-spatial WM as a mediator of the relationship between teacher-rated inattention and boys' math addition scores one year later.**

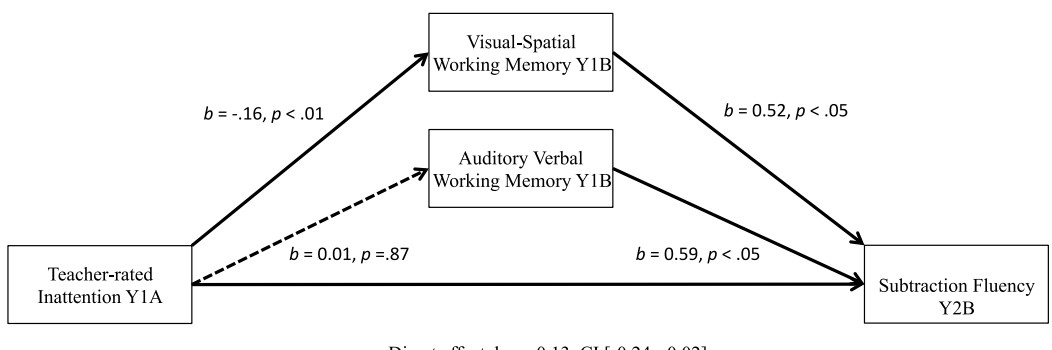

**Figure 2 Significant moderated mediation: visual-spatial WM as a mediator of the relationship between teacher-rated inattention and boys' math subtraction scores one year later.**

$p < .01$) at Year 1B, and children with higher visual-spatial WM scores had stronger scores on addition fluency outcomes at Year 2B (visual-spatial WM: $b = 0.59, p < .05$). A bias-corrected bootstrap confidence interval (BCa CI) for the conditional indirect effect for boys ($ab = -0.06$) based on 10,000 bootstrap samples was entirely below zero ($-0.13$, $-0.01$), therefore is significant. There was also evidence that teacher-rated inattention influenced addition fluency scores the following year independently of its effect on WM (addition $c' = -0.28, p < .001$). The overall model accounts for 59% of the variance for math addition scores at Year 2 ($R^2 = 0.59, p < .001$). For boys, WM and teacher-rated inattention account for an additional 4.5% of the variance in Year 2B addition scores significantly over and above Year 1A addition scores, parent education and age.

Results were similar for subtraction fluency outcomes (see Fig. 2). The only difference was that children with higher visual-spatial and auditory-verbal scores had stronger math subtraction outcomes at the end of Year 2B (visual-spatial WM: $b = 0.52, p < .05$, auditory-verbal WM: $b = 0.59, p < .05$). The overall model accounts 55% of the variance

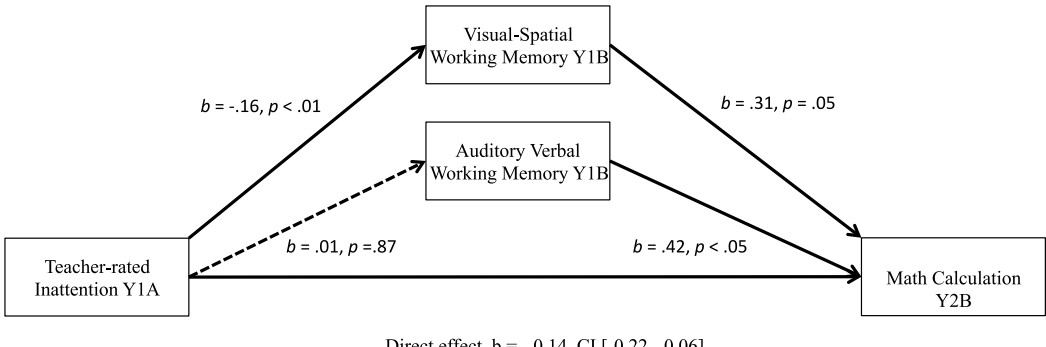

Direct effect, b = -0.14, CI [-0.22, -0.06]

Conditional indirect effect of visual-spatial WM for boys, b = -0.03 CI [-0.07, -0.00]

**Figure 3 Significant moderated mediation: visual-spatial WM as a mediator of the relationship between teacher-rated inattention and boys' math calculation scores one year later.**

for math subtraction scores at Year 2 ($R^2 = 0.55, p < .001$). For boys' subtraction scores, 11.4% of the variance is accounted for by WM and teacher-rated inattention, which is substantially larger than variance accounted for when looking at addition fluency outcomes.

The proposed model was also significant for boys' math calculation outcomes at Year 2B. The conditional indirect effect was significant, $b = -0.03$, BCa CI $[-0.07, -0.00]$, as was the direct effect, $b = -0.14$, BCa CI $[-0.22, -0.06]$, see Fig. 3. Overall, this model accounted for 53% of the variance for Year 2B math calculation scores ($R^2 = 0.53, p < .001$), with teacher-rated inattention and WM accounting for an extra 10.5% of the variance over and above Year 1A calculation scores, parent education and age.

The BCa CIs for all three models passed through zero for girls, thus the mediation through WM was not significant for girls' addition fluency outcomes (visual-spatial WM: $[-0.07, 0.01]$; auditory-verbal WM: $[-0.05, 0.02]$), subtraction fluency outcomes (visual-spatial WM: $[-0.06, 0.01]$; auditory-verbal WM: $[-0.07, 0.00]$) or math calculation outcomes (visual-spatial WM: $[-0.04, 0.01]$; auditory-verbal WM: $[-0.04, 0.00]$).

A study with a comparable design found that with an adolescent sample, the full model accounted for 40% of the variance in math outcomes (*Rogers et al., 2011*), which is somewhat less but still comparable to the variance accounted for in our elementary school-aged sample (59%, 55% and 53% for math addition, subtraction and calculation, respectively). Thus far in mediation research, options for calculating overall effect size are limited to simple mediation models without covariates (*Preacher & Kelley, 2011*). Preacher and Kelley outline the difficulty with classic effect size measures, as they do not fit with indirect effects; the product of two regression coefficients. The most robust effect size measure for indirect effects to date is Preacher & Kelley's Kappa-squared ($K^2$; *Hayes, 2013*; *Preacher & Kelley, 2011*). Therefore, recognizing that this is of limited applicability to our full model that includes covariates, we calculated the $K^2$ for each significant model, that is, for boys, using visual-spatial WM at Year 1B as a mediator, teacher-rated inattention at Year 1A as the predictor, and math scores at Year 2B as outcome variables. Results indicate that for math addition, $K^2 = .06$, BCa CI $[0.01, 0.14]$, for math subtraction $K^2 = .04$, BCa CI $[0.00,$

0.13], and for math calculation $K^2 = .13$, BCa CI [0.06, 0.21]. This can be interpreted as the indirect effect being about 13% of the maximum value that it could have been for calculation outcomes, which is between a medium and large effect (*Field, 2013*; *Preacher & Kelley, 2011*). According to *Field (2013)*, this is a 'reasonable' size for psychological science. The effect sizes for the models with math addition and subtraction fluency as outcome variables are between small and medium effects, with the indirect effects being 4% and 6% of the maximum value that could have been accounted for in the model.

### Reading outcomes

When predicting reading outcomes, although approaching significance, there was no significant direct relationship between teacher-rated inattention and reading fluency ($b = -.27, p = .055$) or word reading ($b = -0.02, p = .54$) scores one year later (see Figs. S1 and S2). There were also no significant mediation effects for reading fluency (visual-spatial WM BCa CI [−0.13, 0.06], auditory-verbal WM BCa CI [−0.16, 0.04]) or word reading (visual-spatial WM BCa CI [−0.02, 0.04], auditory-verbal WM BCa CI [−0.00, 0.02]). Significant predictors of reading fluency at Year 2B were auditory-verbal WM at Year 1B ($b = 1.63, p < .05$), parental education ($b = 3.10, p < .01$), and Year 1A reading fluency scores ($b = 0.76, p < .001$). The only significant predictors of word reading were Year 1A word reading scores and age ($b = 0.78, p < .001, b = 1.88, p < .001$).

To investigate this same model using WM scores at Year 2B, which allows us to control for previous levels of WM at Year 1B (which we cannot do when using Year 1B scores as the mediating variable), we modeled teacher-rated inattention at Year 1A as the independent variable, academic outcomes at Year 2B as outcome variables with WM at Year 2B as a potential mediator. Results replicate the first model in that visual-spatial WM was a significant mediator of the relationship between teacher-rated inattention and math calculation, the confidence interval for the indirect effect for boys was entirely below zero (BCa CI [−0.10, −0.01]). However, WM was not a significant mediator for math CBM addition (BCa CI [−0.10, 0.03]) and subtraction (BCa CI [−0.09, 0.01]), reading fluency (BCa CI [−0.10, 0.12]) or word reading (BCa CI [−0.03, 0.03]). Results from this model need be interpreted with caution, as WM at Year 2B was collected at the same time point as the outcome variables, thus this model is subject to issues with reverse causation.

A reverse model was conducted in order to confirm directionality of the predictor and mediating variables. No mediation models were significant when reversing the role of mediator and independent variable, with Year 1B WM modeled as the independent variable, Year 2A teacher-rated attention as the mediating variable, Year 2B academic variables as outcomes, sex as a moderator and age, parental education, Year 1A academic scores, and Year 1A attention scores as covariates.

## DISCUSSION

The present study contributes to our understanding of the longitudinal relationships between classroom inattention, WM, and math and reading outcomes in a community sample of elementary school children. We hypothesized that inattention would directly and indirectly influence math outcomes through auditory-verbal and visual-spatial WM,

and influence reading outcomes through auditory-verbal WM. Using OLS regression based mediation analyses, we found support for a model in which children's classroom inattention, as rated by teachers at the beginning of the school year, was indirectly associated with math outcomes one year later through visual-spatial WM, but only for boys. There was also a significant direct association between teacher rated inattention and all measured math outcomes the following year. These findings were consistent with our first hypothesis; the proposed model held for math CBM addition and subtraction fluency scores as well as math calculation outcomes.

Our second hypothesis, that auditory-verbal WM would play a role in math outcomes, was not confirmed in the current study. Explanations for this finding based on the literature are discussed below.

These new findings raise interesting questions about sex differences in the role of visual-spatial WM on different aspects of math skill development, and about the role of auditory-verbal WM across academic outcomes.

One interesting result that came out of initial analyses is that there was no significant correlation between auditory-verbal and visual-spatial WM measures in this sample. Factor-analytic and neuroimaging studies suggest that the two aspects of WM are distinct domains that overlap, thus proposing a domain-distinct model of WM (*Alloway & Alloway, 2013*; *Alloway, Gathercole & Pickering, 2006*; *Fassbender & Schweitzer, 2006*). One explanation for the lack of correlation could be that our visual-spatial measure taps into storage, while our auditory-verbal measure taps into both storage and processing components of WM; again, related but separable components (*Alloway, Gathercole & Pickering, 2006*). However, this is unlikely to be the explanation in our sample, because there is also no significant correlation between the two measures of storage only (visual-spatial storage and auditory-verbal storage, $r = .12$, $p = .09$). Furthermore, a lifespan study provided support that working memory skills are not driven by differences in function (storage versus manipulation) but by domain differences (*Alloway & Alloway, 2013*). This finding is further supported by the current study that suggests domain differences in a sample of children, representative of the full range of inattentive behaviours in a classroom setting.

The main finding that visual-spatial WM, measuring storage only, was a significant mediator of the relationship between teacher-rated inattention and both math fluency and math calculation outcomes is consistent with expectations based on current literature. The visual-spatial domain of WM is consistently linked to math outcomes, both when the measure includes manipulation demands and short-term storage only (*Alloway & Passolunghi, 2011*; *Bull, Espy & Wiebe, 2008*; *Li & Geary, 2013*; *Rogers et al., 2011*). However, there were differences in the effect size between the two math domains; with successively larger effect sizes for addition fluency, subtraction fluency and calculation, with the calculation effect size reaching a medium to large effect ($K^2 = .13$). The addition and subtraction fluency tests and the WJ-IIIACH math calculation test all require basic math fact skill, and the calculation subtest builds from these basic skills to include procedural knowledge and higher processing demands (*Fuchs et al., 2006*). There are clear differences between these two measures that may account for the differential influence of WM in
terms of effect size and variance accounted for by inattention and WM. Taken together, the robust effect size for calculation scores, and the fact that this model remained significant when controlling for earlier WM (Year 1B) scores, but did not remain significant for fluency scores, indicates that across time, WM appears to play a more significant role as a mediator between inattention and higher-level math calculation skills than for math fluency skills. As elementary school children move from using counting based methods for solving math facts, to fluent memory-based retrieval, there is a parallel shift from activation in the fronto-parietal WM systems, to increased hippocampal activation (*Qin et al., 2014*). Therefore, it appears that demands on WM for solving math facts are lessened across the early school years, whereas for math calculation tasks that require online processing it is hypothesized that WM load remains high (*Geary, 1994*). It follows to reason that the influences of inattention on higher-level math outcomes, which require more attention to algorithm and less reliance on fluent retrieval, are partially accounted for by visual-spatial WM. Furthermore, our results that differentiate math fluency from higher-level calculation, with similar effects of visual-spatial WM but different magnitude, can be considered in the context of genetic studies which provide evidence that math fluency is a distinct construct from other domains of math (*Petrill et al., 2012*).

Another unexpected finding was related to sex differences, in that visual-spatial WM was a significant mediator for boys but not for girls. This was not specifically addressed in an initial hypothesis, but rather came out of analyses that demonstrated sex differences, therefore leading to the examination of sex as a moderator. To our knowledge, this is the first study to examine sex differences on the role of WM in the relationship between teacher-rated inattention and academic outcomes in the form of standardized achievement. Two studies that looked at variance accounted for by inattention and WM in predicting academic outcomes did not report an assessment of sex differences, outside of the equal distribution of males/females between ADHD and non ADHD groups (*Rennie, Beebe-Frankenberger & Swanson, 2014*; *Rogers et al., 2011*). The current results suggest that although sex differences were not found on visual-spatial WM, this construct plays an important role as a mediator between classroom inattention and math outcomes for boys. One possible explanation is through a line of research with adults, which suggests that spatial numerical associations may be represented differently between males and females (*Bull, Cleland & Mitchell, 2012*). Bull and colleagues hypothesize that men may rely more on spatial representations of number, thus providing a theory of why visual-spatial WM played a mediating role in boys' math outcomes. The study was not replicated with children, however, a study with 8th grade students found that boys' scores on mental rotation predicted math achievement, but this was not found for girls (*Ganley & Vasilyeva, 2011*). This may suggest that in this higher level of 8th grade math, girls rely less on spatial reasoning for math problem solving than do boys. Future research might examine math anxiety as an additional mediator in this model. Studies have found that math anxiety of female-teachers relates to their female students' math performance via endorsement of sex stereotypes of who is good at math (*Beilock et al., 2010*). Therefore it could be that for girls,

math anxiety may play a role such that lower performance is not due to visual-spatial WM difficulties, as it is for boys.

Another finding related to sex differences is differential scores for boys and girls on our measure of teacher-rated inattention. In our mediation model, the direct relationship between inattention and math outcomes was significant for both boys and girls. However, our results show higher levels of inattention for boys, which is consistent with existing research. It has been widely reported that boys have higher levels of inattention as rated by teachers and parents, at least for those who fit diagnostic criteria for ADHD (for a review see *Gershon, 2002*). However conclusions regarding gender differences in symptoms of inattention are somewhat equivocal, depending on sample (*Biederman et al., 2005*). Of particular relevance to the current study, *Ramtekkar et al. (2010)* found a similar pattern to our results, using the SWAN scale in a community sample of children aged 7–12, with girls showing stronger levels of attention than boys.

In terms of sex differences on WM tasks, the current results are in line with findings in clinical ADHD populations (for example *Castellanos et al., 2000*; *Rucklidge & Tannock, 2002*). However, the evidence for sex differences in the general population, on visual-spatial WM is mixed. A body of literature provides consistent findings that males outperform females in visual-spatial rotation tasks, which involve short-term storage as well as transformation (*Masters & Sanders, 1993*). A study with children found that males outperformed females on an abstract visual-memory task and a memory for location task, while females outperformed males on two verbal tasks (*Modesto-Lowe, Yelunina & Hanjan, 2011*). However, similar to our results, there were no sex differences on the visual-sequential memory task (most similar to the WRAML finger windows task in the current study) or the digit span tasks (*Modesto-Lowe, Yelunina & Hanjan, 2011*). Similarly, a study with high-school students found no sex differences on a visual-spatial storage task, but did find that males performed better than females on visual-spatial WM tasks that required processing (*Kaufman, 2007*). Therefore, the current study provides evidence that boys and girls perform equally on the WM tasks included in this study, when examining a community sample of elementary-school aged children.

The finding that auditory-verbal WM did not significantly mediate the relationship between teacher-rated inattention and academic outcomes makes sense, given the mean age of our participants and the nature of the math outcome measures. Conflicting results in the role of auditory-verbal WM in math outcomes may be due to the use of a math composite score in previous studies, that included higher-level math and problem-solving skills that are more strongly associated with verbal WM and executive skills (*Passolunghi & Siegel, 2004*; *Rogers et al., 2011*; *Swanson, 2011*). In addition, although our sample size did not allow for separate mediation analyses within each grade, differences in relative contribution of auditory-verbal WM to math fluency skills between grades were found in the cited studies. Therefore, another possibility is that the children in our sample are young (mean age is 7.67) and may rely mostly on visual-spatial WM to process information at this stage, not having gone through the developmental shift toward relying more on auditory-verbal WM for information processing (*Fastenau, Conant & Lauer, 1998*;

*Raghubar, Barnes & Hecht, 2010*). Another consideration is the presentation format of math problems. It is possible the presentation of problems in a vertical format influenced our results (although 5/45 questions in the math calculation task were horizontal), in that this presentation format recruits more visual-spatial WM resources than math problems that are presented in horizontal format (*Trbovich & LeFevre, 2003*).

A second main hypothesis in the current study was that inattention would indirectly influence word reading and oral reading fluency outcomes, through auditory-verbal WM. We did not find such an indirect effect, and interestingly, the direct effect of teacher-rated inattention at Year 1A on Year 2B reading fluency and word reading scores was also not significant, while controlling for Year 1A reading fluency scores. In the context of other studies in which inattention is a predictor of reading fluency (for example *Pham, 2013*), it is important to note that in the current study, inattention and WM were modeled along with covariates, including parental education, age and Year 1A reading fluency scores, which all significantly predicted reading fluency scores at Year 2B. Examination of this sample within a one year time frame indicated a significant relationship between inattention at Year 1A and reading fluency at Year 1A (S Gray et al., 2014, unpublished data). However, contrary to these findings, inattention does not appear to play a significant role in predicting reading scores one year later independent of its association with reading fluency at Year 1A. This highlights the strong stability of reading fluency across the elementary school grades. However, auditory-verbal WM was a significant predictor for reading fluency. Thus, although not found to play the hypothesized mediating role, auditory-verbal WM is positioned as an important factor in the development of reading fluency across the elementary school years. These results are consistent with the findings of *Li & Geary (2013)*, who also found that visual-spatial WM was not a predictor of reading outcomes, but that gains in visual-spatial WM were associated with stronger math scores at the end of elementary school.

Although outside the scope of this paper, future studies could seek to account for other mediators within the relationship between inattention and academic outcomes, in addition to the aforementioned construct of math anxiety. The current study focused on WM, however other cognitive functions, such as processing speed and naming speed are important to consider in relation to academic fluency (*Fuchs et al., 2008*; *Jacobson et al., 2011*; *Martinussen, Grimbos & Ferrari, 2014*). WM and processing speed are highly related in the early years of development. However, recent findings suggest that these two constructs independently predict academic fluency as children move into the elementary-school aged years (*Clark et al., 2014*; *Jacobson et al., 2011*). Therefore, future studies might examine the role of processing speed as well as WM in the relationship between classroom inattention and academic fluency, while also controlling for more proximal indicators such as phonemic awareness. Other more distal mediators, such as teacher instructional supports, and parent factors such as support with homework, may be important to consider as mediating variables in community samples (*Daley & Birchwood, 2010*; *Langberg et al., 2011*).

Strengths of this study include the large sample size, as well as the longitudinal design, in which the predictor is collected before the mediating variables, and the outcome variables are collected one year later. Separation of WM domains and academic skill outcome

variables allowed for a more specific understanding of these relationships across two school years.

There are some limitations to consider in the design and interpretation of this study. One limitation includes our narrow measures for WM, as we did not have a visual-spatial WM measure that required processing or manipulation, the WRAML measure only taps into short-term storage. Although measures used in the current study are highly reliable, future studies might employ a variety of WM measures for each domain, providing the ability to create a comprehensive composite for each domain. A previous study found that visual-spatial storage was a strong predictor of math achievement early in the first grade, but by age 8, a visual-spatial measure that required manipulation also predicted math outcomes (*Bull, Espy & Wiebe, 2008*). It could be that the importance of visual-spatial WM was underestimated by not including a measure that would include all components of visual-spatial WM. Indeed, one study found that the manipulation component of visual-spatial WM did contribute additional variance to math outcomes above the contributions of visual-spatial short-term storage (*Geary, Hoard & Nugent, 2012*).

Another limitation in our study design is that practical considerations limited data collection time points, such that our mediating variable was collected at 2 time points across two years, whereas the outcome variables were collected at 4 time points. It would have been ideal to have baseline WM measures, however, time allotted by the school for each testing session as well as date restrictions did not permit for collecting cognitive measures for the full sample of 524.

Strengthening the confidence in outcomes is the fact that when reverse modeling WM and inattention, where WM is placed at the predictor, and inattention as the mediator, WM measures at Year 1 do not predict levels of inattention in Year 2. Our outcomes regarding visual-spatial WM are afforded more confidence, as we were able to run the analysis with WM at Year 2 as the mediating variable, thus accounting for the influence of prior WM scores (at Year 1). This model continued to reach significance for calculation outcomes, however this model is interpreted with caution, as WM measures at Year 2 were collected at the same time point as the academic outcome measures. Therefore, although further evidence is needed to substantiate the developmental directionality between inattention and WM, these findings add to our knowledge about longitudinal predictors of academic outcomes in elementary school children and further specify the nature of the relationship between inattention, WM and academic outcomes across elementary school, for typically developing children.

## CONCLUSIONS

Findings extend previous research and confirm and replicate the body of literature that positions behavioral inattention as a robust predictor of later math achievement, further specifying that this relationship is robust for arithmetic fluency and algorithmic computation, in typically developing elementary school children. Further, findings add new information about the role WM plays for boys and girls in the relationship between inattention and math and reading outcomes across two school years. Contrary to results

from cross-sectional studies, our findings provide evidence that after controlling for initial reading fluency scores, auditory-verbal WM is a more robust predictor of reading fluency across two school years, as compared to teacher-rated inattention, which did not predict growth in reading fluency beyond the contribution of Year 1A inattention.

Although math fluency shares significant variance with inattention and both domains of WM (S Gray et al., 2014, unpublished data), the current study provides evidence that boys' classroom inattention (as rated by teachers) directly influences their math fact fluency and math calculation scores across time, controlling for age, parental education and math scores at Year 1. Main findings emphasize that for boys, inattention has a direct effect as well as an indirect influence on math fluency and calculation skills through visual-spatial WM in the elementary school grades.

### Funding

Funding for this project was provided by the Canada Research Chair Program (RT) and the Joseph-Armand Bombardier CGS Doctoral Scholarship (SG) SSHRC #410-2008-1052. The funders had no role in study design, data collection and analysis, decision to publish, or preparation of the manuscript.

### Grant Disclosures

The following grant information was disclosed by the authors:
Canada Research Chair Program (RT).
Joseph-Armand Bombardier CGS Doctoral Scholarship (SG).
SSHRC: #410-2008-1052.

### Competing Interests

Dr. Tannock is on the advisory board for Eli Lilly, a consultant for Purdue Pharma Canada, and in 2010 participated in an ADHD meeting sponsored by Janssen-Cilag.

### Author Contributions

- Sarah A. Gray performed the experiments, analyzed the data, wrote the paper, prepared figures and/or tables, reviewed drafts of the paper.
- Maria Rogers and Rhonda Martinussen conceived and designed the experiments, reviewed drafts of the paper.
- Rosemary Tannock conceived and designed the experiments, performed the experiments, reviewed drafts of the paper.

### Human Ethics

The following information was supplied relating to ethical approvals (i.e., approving body and any reference numbers):

Institutional Review Board of the Hospital for Sick Children, Toronto, Ontario, Canada.

Sickkids REB #1000013136, "Inattentive behaviors and cognition as predictors of later academic outcomes" (PI: Rosemary Tannock).

## Supplemental Information

Supplemental information for this article can be found online at http://dx.doi.org/10.7717/peerj.939#supplemental-information.

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
