# Peer review of "Longitudinal relations among inattention, working memory, and academic achievement: testing mediation and the moderating role of gender"

_PeerJ, doi:10.7717/peerj.939_

## Round 0.1 · original submission · Major Revisions

Dear Dr Gray and Professor Tannock,

Thank you for you submission of the manuscript titled “A longitudinal study of potential mediators of the relationship between inattention and academic achievement in a community sample of elementary school children”. And thank you for your patience while I found reviewers.

I have received two reviews now and both reviewers see clear merit in the work and raise useful points that deserve consideration. I’ll summarise the points I feel are most critical, including some comments of my own. Overall, tackling these suggestions feels like major revisions to me but I do see them as useful to the evaluation of the work.

I have numbered and lettered these comments with the aim of clarifying where a separate response is relevant. Please only follow the lettering where this is intuitive to you.

1. Working Memory versus Short-Term Memory

As noted by Dr St Clair Thompson and in my own reading, the visual-spatial task only includes a forward span element, which is better described as short-term memory (STM) rather than working memory (WM). In contrast, the auditory task includes forwards and backwards elements, combining STM and WM. This is most likely just a question of renaming this variable throughout; however, as the auditory task includes both STM and WM, the overall theme becomes less elegant to summarise. Depending on how you choose to address this, it may constitute significant changes.

In addition, I was surprised that the visual-spatial and auditory tasks weren’t significantly correlated considering the theoretical overlap with respect to the central executive. It may be the case that this lack of correlation relates to the absence of WM for the visual-spatial task. Considering the differences between STM and WM may resolve this. Having said that, whilst it surprises me, I have no issue with it being the case. I just wanted to flag it for consideration and potential discussion.

2. Reference to the Gray et al. submitted manuscript

There seem to be details in your submitted manuscript which have a bearing on the current manuscript such as sample descriptives (Dr Boyes #6 maybe #7). I also note on page 7:

“As has been reported, the only significant difference between male and female participants was that females were more likely to have an informant (92.3% of informants were mothers) with less than high school education.”

Presumably this would be clear if we had access to the Gray et al. submitted article but as we don’t, a significant difference is mentioned here but only one side of this difference is described. I think the other side needs to be covered to give a complete picture here.
Additional information in the current manuscript would be ideal, however, considering publishing the submitted paper as a PeerJ Pre-print and referring to it rather than the submission may be a reasonable alternative.

3. Speed of processing and data reduction

a) Dr St Clair-Thompson notes, under Experimental design, that the math tasks may include a speed of processing component. I agree with this and her suggestions for discussion and/or acknowledgement of this.

b) In my own reading I also noted, in reaction to the correlations (table and page 12/13), that here’s a lot of overlap between the math tasks as well as the two reading variables (some correlations of .8 or more). Some form of data reduction (e.g., factor analysis) may be worth considering to clearly delineate the math skills examined (i.e., whether the qualitative differences between the tasks are reflected in the data). In addition, this might indicate a speed-related factor, which could then be excluded or considered in the analysis.

c) This can also be related to Dr Boyes comments #3, 4, and 5. Outlining the anticipated relationships would be helpful to the paper and, if data reduction were to be undertaken, it would have a bearing on interpreting these outcomes.

4. Additional task/scale information

As noted by Dr Boyes (#8), the reliabilities of the tasks/scales implemented are an important inclusion. The quality of the measures is critical for interpreting the quality of the results. Further information about the number of items, e.g., for the SWAN, would be helpful to include. It is possible these details are in the submitted manuscript but as noted above under point number 2, this can’t currently be accessed.

5. Months and years

I too found the reference to November and April in by study years confusing (Dr Boyes #9). Feel free to address this however you see fit but to offer a suggestion, referring to school terms (i.e., November = term 1 or 2 and April = term 3 or 4 – whatever the terms actually are) might work. I think this would give a clearer description of the sequence of events for those less familiar with the Canadian system.

6. Meditation analyses

a) As Dr St Clair-Thompson notes (under Validity of the findings), I think it would be helpful to have a further explanation of why small values are significant and large values aren’t.
b) Dr Boyes (#11) requests the overall variation accounted for by each model. This would be a useful inclusion.
c) I think it’s good practice to report the details of non-significant statistics in some form. These come in handy for meta-analyses and replications. Consider adding all of these details as supplementary information.

7. Task/scale results

On a final note from myself, I thought it would be helpful to have a summary (i.e., descriptives) of the sample task/scale scores overall. This might be useful for the discussion – was the visual task more difficulty that the auditory? Or did this particular sample find it harder, hence the relationships. These could be included as an addition to the correlation table (i.e., and extra column or two).

I particularly feel that longitudinal data on this scale is important to be included in the scientific literature. I hope that the comments feel constructive and addressing them will be a good use of your time.

·

Basic reporting

All comments are made in the general comments to the author section

Experimental design

All comments are made in the general comments to the author section

Validity of the findings

All comments are made in the general comments to the author section

Additional comments

This is an interesting paper examining the relationship between inattention and maths and reading ability, and examines working memory (WM) as a potential mediator. Findings demonstrate that visual-spatial (but not auditory-verbal) WM mediates the relationship between inattention and maths ability, but not for reading ability. Inattention was also directly associated with maths ability. Strengths of the manuscript include the collection of data from multiple time points and the use of contemporary statistical methods for testing indirect effects. However, I have a number of questions and comments that I believe will improve the manuscript. These are itemised below.

Title

1) Whilst the study does examine two mediators, these are both dimensions of WM. By stating the paper is a “longitudinal study of potential mediators”, the title raises an expectation that more constructs would be examined? I think the title should specify that the study focuses on WM as a potential mediator.

Introduction

2) The acronym WM is used on page 2 without the full label being used prior.

3) On page 3 WM is implicated as important in math word problems, but from the description algorithmic computation (which requires the ability to follow procedural steps) is WM not important in this as well?

4) Relatedly, is there evidence for a role for WM in reading and can this be used as an additional rationale for examining WM as a potential mediator? This is touched upon, but I think a bit more information would be useful.

5) Additionally, given that WM is hypothesized to mediate the relationship between inattention and academic performance, I would also like to see some discussion of why inattention should predict poorer WM?

Materials and Methods

6) The sample is not described in great detail. The authors state that this is because detailed data on the sample has been previously described. However, this paper is not published (the reference section says it is submitted) and this makes it difficult to evaluate the sampling procedures.

7) All participants were in mainstream English or French classrooms. Were there any differences on the variables of interest between children in English and French classrooms, and if so should language be adjusted for in the analyses? Were assessments conducted in English, French, or both depending on the school/child?

8) No information regarding the reliability or validity of the SWAN Scale is provided. In particular, what was the reliability in the current sample? Similarly, some information of the reliability and validity of the maths, reading, and WM measures is needed.

9) I was a little confused about the temporal ordering of the assessments. If I have it correct, there were 4 assessment points: 1) April Year 1, 2) November Year 1, 3) April Year 2, 4) November Year 2. Questionnaire measures of inattention and maths and reading ability were collected in November of Year 1 and Year 2. Working memory measures were collected in April of Year 1 and Year 2?

This is confusing, but may just be Australia (or Southern Hemisphere) specific - as here the academic year follows the calender year. Based on the calendar year, the WM assessments preceded the questionnaire measures? I assume therefore that Year 1 ran from November to November (and this would fit with the Year 1 A and Year 1 B labels etc), and the same for Year 2? If this is the case it needs to be made a bit more explicit in the manuscript. Perhaps removing the months and simply saying that there were 2 assessment points each year (or using T1-T4) would help with this?

Results

10) See my comment 7 – are there relationships between language and variables of interest, and if so should language be controlled for?

11) How much of the variance in the outcome was accounted for in each model?

12) The non-significant associations in Figure 1 are not illustrated – these should be identified by dashed lines

Discussion

13) It is suggested that future studies could examine other potential mediators, including parent and individual factors. It would be good to see some specific examples of parent and individual factors specified?

14) The strengths of the study are flagged in the discussion, but it would be good to see some thought given to potential limitations of the study?

General

15) Very minor comment, but I think that both the introduction and discussion sections would benefit from an additional level of subheading, which would allow separate sections related to maths ability and reading ability.

Thank you for the opportunity to review this manuscript. I do think the paper is interesting and has the potential to add to the understanding of factors associated with maths and reading ability. I hope that you and the authors find my comments useful.

Mark Boyes

Lecturer (Research Fellow)
School of Psychology and Speech Pathology
Curtin University
Perth, Western Australia

·

Basic reporting

In general the article is well-written and provides a thorough overview of previous literature on this topic. However, the introduction could say a bit more about behavioural inattention. What behaviours indicate inattention, how is it measured, and how is it distinct from cognitive inattention?
The article is also well-structured with clear Tables and Figures. A minor point, however, is that there are several uses of "et al." when authors surnames should have been provided (i.e. the first time a study is cited).

Experimental design

The article is within the scope of the journal. It also clearly outlines the research question and appropriate hypotheses. The methods used and statistical analyses are also suitable.
I note, however, that the visual-spatial working memory measure was a measure involving storage with no supplementary processing. Therefore the term visual-spatial short-term memory may be more appropriate to use than working memory.
I also note that the maths tasks (particularly those for addition and subtraction) were conducted within a very short duration of only 2 minutes. It is likely that speed of processing is therefore important. The article would benefit from a discussion, or at least acknowledgement, of this issue.

Validity of the findings

The data reported appear to be robust and statistically sound. However, as a non-expert in mediation analysis one aspect of the results is somewhat confusing. In each of the Figures 1,2, and 3, the value of visual-spatial working memory as mediating the inattention- maths relationship is very small (e.g. -0.06.). This is noted and discussed as being significant. Elsewhere in the Figures, larger values (e.g. .10 in Figure 1, and .53 in Figure 2) are noted as non-significant. I think there needs to be some explanation of these statistics, and most importantly a discussion of effect size.

Additional comments

There is an inconsistency between the abstract and the method- the percentage of female participants is reported as 50% and then 49%. I also struggled to understand the section in assessment of classroom attention beginning "This scale avoids the psychometric flaws of many.....". Further explanation is required.

---

## Round 0.2 · Minor Revisions

Dear Dr Gray and colleagues,

Thank you for taking the time to address the comments to the original manuscript. I agree with the reviewers that you’ve done a great job answering these concerns.

Dr Boyes has some relevant questions regarding the inclusion of sex what would be useful to consider. And I have a few minor notes below – two just as confirmation for you and a couple that constitute adjustments for your consideration.

I note that:
Regarding editor point 6:
“If the journal would prefer us to report standardized coefficients, we can standardize the variables before analysis, as PROCESS does not produce standardized coefficients – please let us know if this would be preferable.”
I’m not aware that the journal has a policy on this so unstandardized is fine.

Mark Boyes question 12) dashed lines
The resolution of the diagrams looks good to me

1. But I did note in Figure 3 that the path between Visual Spatial Working Memory Y1B and Math Calculation Y2B has p = .05 and is not dashed. I don’t have an issue with this but wanted to flag it in case it was meant to be less than .05 – some sticklers worry about such things…

2. I had some minor points about Table 2.
a) Perhaps the sample size could be delineated by gender; that is, rather than n = 204 the title could read: ‘...for Girls (n = 101) and Boys (n = 103)’
b) I wondered whether t-tests might be most appropriate for the two-group comparisons
c) And it’d be good to have effect size of some description to accompany to the inferential statistics

3. Working Memory (WM) in the abstract
Related to editor point number 1, surrounding concerns that some ‘light’ readers might not grasp this subtly and misrepresent the WM task in their heads or when referencing the current work, could some reference be made to the nature of this task within the abstract? I appreciate that this may be difficult to do in an abstract, and may be confusing for the reader so this is for your consideration – not something to lose sleep over!

·

Basic reporting

All comments are made in the general comments to the author section

Experimental design

All comments are made in the general comments to the author section

Validity of the findings

All comments are made in the general comments to the author section

Additional comments

The authors have done a thorough job in addressing all of my previous comments and I think the paper is much improved.

However, the authors have made an additional change to the manuscript (i.e. incorporating sex as a moderator) and I do have some comments regarding this that I think should be addressed. These are relatively minor comments, but I feel that addressing point 1 would clarify why gender might be conceptualised as a moderator, and addressing point 2 would just clarify the analytic approach.

1) In the results these analyses seem to come out of the blue? The introduction doesn't really make a case for why gender should be a moderator, and this isn't mentioned at all in the aims/hypotheses section? I agree with the authors that conceptualising gender as a potential moderator is interesting; however, I think this needs to be incorporated into the introduction and study aims. There is some relevant discussion (pages 24-25), but I think it is important to bring this into the paper earlier.

2) PROCESS has a number of potential models for moderated mediation, which specify which individual path(s) in the mediational model are actually moderated (or directly specifies all paths in the case of model 59) - see models 7, 8, 14, 15, 58, and 59 in Hayes 2013 for some examples. Was gender allowed to moderate all potential paths or was it limited to certain paths in the model? Also where the relevant interaction terms significant? I have seen cases where the mediation model was significant at only certain levels of the moderator, but the relevant interaction term was not significant? Given the authors are providing other supplementary information, perhaps it might be worth including the regression tables in supplementary materials?

Thank you for the opportunity to review this revision and I hope the authors find my comments helpful.

Mark Boyes
Lecturer (Research Fellow)
School of Psychology and Speech Pathology
Curtin University
Perth, Western Australia

·

Basic reporting

No comments.

Experimental design

The information which has been added to the manuscript has helped to clarify some of the previous issues with experimental design. I was pleased to see that the authors have added information about reliability of the measures, about the distinction between short-term memory and working memory (or storage versus storage and processing) tasks, and also about the role of processing speed.

Validity of the findings

Again, the revisions have resolved any issues with the validity of findings. The revised manuscript presents findings clearly and makes appropriate conclusions, and now provides sufficient discussion of relevant points.

Additional comments

I am satisfied that the authors have addressed each of the comments made by the reviewers and by the Editor. The revisions have resulted in a much improved manuscript. I am happy to recommend that the manuscript is now accepted for publication.

---

## Round 0.3 · accepted · Accept

Dear Dr Gray and colleagues,

Thanks again for responding to these minor comments. I'm happy with your responses and don't feel that it's necessary for the paper to be returned to the reviewers.

I would ask that when you deal with PeerJ's production, you should include the regression tables (as mentioned by Mark Boyes comment 2) as supplementary materials. As these will only be accessed by interested readers and not part of the main body, it doesn't feel to me that there's a cost to including them. In fact, if anyone is ever interested in these details for a review or meta-analysis, having them readily available through the PeerJ site for the paper will be convenient - and probably save you the hassle of trying to dig it at a later date.